# Microglia Depletion from Primary Glial Cultures Enables to Accurately Address the Immune Response of Astrocytes

**DOI:** 10.3390/biom12050666

**Published:** 2022-05-04

**Authors:** Mariana Van Zeller, Ana M. Sebastião, Cláudia A. Valente

**Affiliations:** 1Instituto de Farmacologia e Neurociências, Faculdade de Medicina, Universidade de Lisboa, 1648-028 Lisboa, Portugal; mariana.campos@medicina.ulisboa.pt (M.V.Z.); anaseb@medicina.ulisboa.pt (A.M.S.); 2Instituto de Medicina Molecular João Lobo Antunes, Faculdade de Medicina, Universidade de Lisboa, 1648-028 Lisboa, Portugal

**Keywords:** mixed glial cultures, astrocytes, microglia, CSF-1R, PLX-3397, C3, PTX3

## Abstract

Astrocytes are the most abundant cells in the CNS parenchyma and play an essential role in several brain functions, such as the fine-tuning of synaptic transmission, glutamate uptake and the modulation of immune responses, among others. Much of the knowledge on the biology of astrocytes has come from the study of rodent primary astrocytic cultures. Usually, the culture is a mixed population of astrocytes and a small proportion of microglia. However, it is critical to have a pure culture of astrocytes if one wants to address their inflammatory response. If present, microglia sense the stimulus, rapidly proliferate and react to it, making it unfeasible to assess the individual responsiveness of astrocytes. Microglia have been efficiently eliminated in vivo through PLX-3397, a colony-stimulating factor-1 receptor (CSF-1R) inhibitor. In this work, the effectiveness of PLX-3397 in eradicating microglia from primary mixed glial cultures was evaluated. We tested three concentrations of PLX-3397—0.2 μM, 1 μM and 5 μM—and addressed its impact on the culture yield and viability of astrocytes. PLX-3397 is highly efficient in eliminating microglia without affecting the viability or response of cultured astrocytes. Thus, these highly enriched monolayers of astrocytes allow for the more accurate study of their immune response in disease conditions.

## 1. Introduction

The brain is separated from the rest of the body by a physical barrier, the blood–brain barrier (BBB), that limits the entrance of cells and compounds into the central nervous system (CNS) parenchyma [1]. Thus, cells that compose the peripheral immune system, such as T and B cells, neutrophils and macrophages, are hindered from participating in most of the immune responses that occur in the CNS [2]. Indeed, the brain and spinal cord have their own immune system, which is composed of glial cells, mainly microglia and astrocytes [2].

Astrocytes derive from the neuroectoderm and are the most abundant cells in the CNS parenchyma, outnumbering neurons, playing a role in the modulation of immune and inflammatory responses in addition to participating in the fine tuning of synaptic transmission, glutamate uptake, synaptogenesis and structural support [3,4]. Astrocytes are also of major importance for the maintenance of the BBB and the clearance of dead cells and cellular debris [5]. In response to damage/insults, these cells undergo a process named astrogliosis, which involves changes in their structure and morphology, alterations in gene expression and the formation of an astrocytic scar to encase necrotic lesions [5,6]. This glial scar is a dense network of long processes that enclose the core of severely damaged sites [7]. Although in physiological circumstances, astrogliosis exerts a protective role [8], in a pathological context, it can be harmful to neurite growth and neuronal regeneration [9,10,11].

Microglia are the first line of defense of the brain’s immune system [11], and due to this immunological function, they are considered the macrophage-like cells of the brain [12]. They are derived from the yolk sac and populate the CNS very early in development, acting mainly as monitors of the diverse brain regions, maintaining homeostasis and assuring the correct development and neuroprotection of the CNS [13,14].

Much of the knowledge of the biology and function of astrocytes arose from the study of primary cultures of rodent astrocytes [15] since these are easy and convenient to settle in the laboratory [16]. There are minor differences in the preparation of these cultures among researchers [17,18], but the basis is the dissociation of cells from the dissected brain region and their plating on culture dishes [19]. The final cultures result in a mixed population of approximately 90% astrocytes, with the remaining being mostly microglia [16]. Although for most applications, this astrocyte yield is adequate, 10% of microglia is too much if one wants to stimulate astrocytes with a noxious stimulus to address their inflammatory response in a disease context. When sensing the stimulus, microglia rapidly proliferate and react against it, thus changing their relative percentage in the culture and making it unfeasible to address the responsiveness of astrocytes on their own [20]. Furthermore, the presence of reactive microglia changes the gene expression of astrocytes. Indeed, Liddelow and colleagues [21] have shown that factors secreted by pro-inflammatory reactive microglia (C1q, IL-1α and TNFα) strongly impact the reactivity of astrocytes. Although the interaction between these cells plays an unquestionable role in maintaining CNS homeostasis and protection, the contribution of each individual population to the modulation and propagation of disease is of great importance for developing novel therapeutic approaches. It is therefore crucial to find a simple and effective method to eliminate microglia from primary mixed glial cultures.

The method used in many studies to increase the yield of cultured astrocytes is the overnight shaking of cultures dishes in an orbital shaker. This method is based on the principle that microglia and oligodendrocytes are loosely attached to culture dishes. Upon shaking, most of them are released from the bottom of the dish to the culture medium and subsequently eliminated by medium exchange [16]. Over the years, several methods, such as the addition of 1-β-d-arabinofuranosylcytosine (Ara-C) [22] or clodronate [15] to mixed glial cultures, have been pursued with the aim of achieving enriched cultures of astrocytes. Recently, magnetic cell sorting [20] and immunopanning [23] were also proposed to limit the microglia presence in astroglial cultures.

Another promising way to eliminate microglia is through the usage of a colony-stimulating factor-1 receptor (CSF-1R) inhibitor [24]. In the CNS, CSF-1R is only expressed by microglia [25], and CSF-1R signaling is essential for the development, proliferation, maturation and survival of these cells [25,26,27].

In this sense, CSF-1R inhibitors—PLX-3397 and PLX-5622—that cross the BBB and rapidly eliminate microglia have been explored [28,29,30,31,32]. Elmore and coworkers [31] reported that treatment of mice with PLX-3397 for 21 days resulted in the elimination of 99% of microglia, which remained absent for as long as the treatment persisted. Additionally, seven days of PLX-5622 treatment was sufficient to deplete 90% of microglia from mouse brains [32]. With both inhibitors, microglia repopulation occurred a few days following drug withdrawal [31,32]. Recently, it was reported that PLX-3397 was also able to deplete microglia (>90%) in organotypic hippocampal slices [33].

As primary cultures of astrocytes are a widely used method to study the function of these cells in both health and disease, this work aimed to evaluate the ability of PLX-3397 to deplete microglia from primary cultures of forebrain astrocytes, in combination with the method of overnight shaking. Our results point to the high efficacy of PLX-3397 in eliminating microglia while reducing the reactivity of cultured astrocytes without affecting their viability. Thus, we provide a novel procedure to attain highly enriched monolayers of resting-like astrocytes, allowing for more precise in vitro studies on the contribution of astrocytes to the neuroinflammatory response within a given disease context.

## 2. Materials and Methods

### 2.1. Animals

This work used newborn Sprague–Dawley rats. Pregnant females were obtained from Charles River Laboratories (Barcelona, Spain). All experiments were conducted respecting the European Union Guidelines (2010/63/EU) and the Portuguese law regarding the protection of animals for scientific purposes. Experiments were approved by the Ethical Committee of the Faculdade de Medicina da Universidade de Lisboa. All efforts were made to minimize animal suffering and to use the minimum number of animals.

### 2.2. Primary Cultures of Astrocytes

Briefly, as depicted in Figure 1, newborn rat pups were sacrificed by decapitation, and the brain was dissected in ice-cold Phosphate-Buffered Saline solution (PBS, 137 mM NaCl, 2.7 mM KCl, 8 mM Na_2_HPO_4_.2H_2_O and 1.5 mM KH_2_PO_4_, pH 7.4). The forebrain was removed under a dissection microscope and collected in supplemented Dulbecco’s Modification of Eagle’s Media (DMEM:F-12, Nutrient Mixture F-12 with GlutaMAX supplement, Gibco, UK) containing 10% Fetal Bovine Serum and 1% Antibiotic/Antimycotic solution (both from Sigma, Ronkonkoma, NY, USA). Next, cells were mechanically dissociated with a serological pipette and filtered through a 230 μm cell strainer. The cell suspension was transferred to a 15 mL Falcon and centrifuged at room temperature (RT) for 10 min at 200 g. The supernatant was discarded, and the pellet was resuspended in DMEM:F-12, dissociated, filtered using a 70 μm cell strainer and centrifuged again as described above. The pellet was resuspended in DMEM:F-12 (15 mL/brain). Cell suspension was plated on poly-D-lysine hydrobromide (PDL, 10 µg/mL, Sigma) coated T75 flasks. After one week, cells were released from the flasks with 1 mL of 0.05% Trypsin-EDTA (Sigma) for 1 min at 37 °C, resuspended in DMEM:F-12 and replated according to the desired assay: (1) for Western blotting, cells were plated in 6-well plates, also previously covered with PDL (10 μg/mL), and (2) for immunocytochemistry and propidium iodide (PI) uptake assays, cells were plated on glass coverslips placed in 24-well plates and previously covered with PDL (25 μg/mL).

All cultures were incubated in a humified atmosphere at 37 °C and 5% CO_2_ and maintained by changing the whole medium twice a week. At 17 days in vitro (DIV), cells were lysed for Western blotting or fixed for immunocytochemistry. PI uptake assays were performed at 10 and 17 DIV. Insult with microglia-derived factors (F) was performed at 17 DIV and lasted 24 h.

### 2.3. Conventional Shake-Off Method

At 7 DIV, mixed glial cultures were agitated overnight at 300 rpm and 37 °C in an orbital shaker to remove loosely attached glial cells, mostly microglia, as described previously [34,35,36]. The supernatant was discarded, and fresh DMEM:F12 was added.

### 2.4. Addition of PLX-3397 to Primary Cultures of Astrocytes

Pexidartinib hydrochloride (PLX-3397) was obtained from MedChemExpress (Middlesex, NJ, USA). PLX-3397 stock solution (100 mM) was prepared in sterile dimethyl sulfoxide (DMSO, Sigma). To study the efficacy of three different concentrations of PLX-3397, namely, 0.2 μM, 1 μM and 5 μM, an intermediate solution of 50 μM (1:2000 dilution from PLX-3397 stock solution) was prepared in culture medium. PLX-3397 was added to the medium in every medium exchange from the replating procedure onwards.

When dissolving a drug in DMSO, one must consider that, above a certain concentration, DMSO can be toxic to cells/tissues. However, the highest PLX-3397 concentration tested, 5 µm, corresponds to 0.005% DMSO, which, according to the literature, is innocuous [37,38]. Thus, cultures not exposed to PLX-3397 were used as controls.

### 2.5. Whole-Cell Lysates and Protein Quantification

Cells were scraped into 120 μL of lysis buffer containing 50 mM Tris pH 8.0, 5 mM ethylenediaminetetraacetic acid (EDTA, Sigma), 150 mM NaCl, 1% nonyl phenoxypolyethoxylethanol (NP-40, Fluka Biochemika, Buchs, Switzerland) and 10% Glycerol (Sigma) and supplemented with protease inhibitors (complete Mini- EDTA-free, Sigma) and 1 mM phenylmethylsulfonyl fluoride (PMSF, Sigma).

Cell suspension was thawed on ice and homogenized by sonication (Sonics & Materials Inc, Newtown, CT, USA). The suspension was incubated at 4 °C with slow agitation for 15 min, followed by centrifugation at 13,000 g for 10 min at 4 °C to remove cell debris. The supernatant was collected and stored at −20 °C until further use.

Total protein was quantified according to the Bio-Rad DC Protein Assay Kit (Bio-Rad, Hercules, CA, USA) in a 96-well flat-bottom plate. Bovine Serum Albumin (BSA, NZYTech, Lisboa, Portugal) was used to prepare a calibration curve (ranging from 0–1 mg/mL). The absorbance was measured at 750 nm in a Microplate Reader TECAN Infinite M200 (TECAN Trading AG, Männedorf, Switzerland).

### 2.6. Western Blotting

Samples (35 μg of total protein per lane) were prepared in sample buffer and denatured for 10 min at 90 °C. Samples and molecular weight marker (MWM, NZYColour Protein Marker II, NZYTech) were electrophoresed on a 12% sodium dodecyl sulfate–polyacrylamide gel (SDS-PAGE) for 2 h at 120 V and electrotransferred to a polyvinylidene fluoride (PVDF) membrane (Immuno-blot^®^ PVDF Membranes for Protein Blotting, Bio-Rad) at a constant current of 350 mA for 150 min. Subsequently, samples were blocked for 1 h with 3% BSA in Tris-Buffered Saline with Tween-20 (TBS-T, 200 mM Tris-HCL pH 7.6, 1.5 M NaCl, 0.1% Tween) and incubated with the primary antibody diluted in blocking solution overnight at 4 °C. The primary antibodies used were rabbit-raised anti-glial fibrillary acidic protein antibody (anti-GFAP, 1:10,000, Sigma, G9269), goat-raised anti-ionized calcium-binding adaptor molecule 1 antibody (anti-Iba-1, 1:500, Abcam, ab5076), goat-raised anti-Complement 3 antibody (anti-C3, 1:500, R&D Systems, AF2655) and mouse-raised anti-Pentraxin-3 antibody (anti-PTX3, 1:500, Santa Cruz, SC-373,951). Membranes were washed several times with TBS-T and incubated at RT for 1 h with mild agitation with the HRP-coupled secondary antibody (anti-mouse, anti-rabbit or anti-goat from Bio-Rad), diluted to 1:5000 in blocking solution. Lastly, the membranes were rinsed and incubated with the ECL Western Blotting Detection System (Perkin Elmer, Waltham, MA, USA).

Proteins were detected in two sets, as molecular weights overlapped. To probe for the second set of proteins, a stripping step was required. For that, after thoroughly washed with TBST, membranes were incubated with Stripping Solution (15 g glycine, 1 g SDS, 10 mL Tween-20, pH 2.2, up to 1 L with H_2_O MQ) for 30 min at RT. After stripping, membranes were thoroughly washed with distilled water and several times with TBST. The protocol proceeded from the blocking step.

Development of signal intensity was visualized in Image Lab software 5.2.1 associated with ChemiDoc XRS^+^ system (Bio-Rad). The integrated intensity of each band was calculated using computer-assisted densitometry analysis with ImageJ software 1.52q. The protein band intensity was normalized to that of GAPDH. The representative image of each protein evaluated was prepared in Image Lab software by merging the chemiluminescence image with the colorimetric image of the MWM.

### 2.7. Immunocytochemistry

Cells were fixed with 4% paraformaldehyde (PFA, Sigma) in PBS for 15 min at RT. PFA residues were washed with freshly prepared 0.1 M Glycine (NZYTech) in PBS for 10 min at RT. Cells were permeabilized with 0.5% Triton X-100 (Sigma) in PBS for 10 min at RT and blocked with PBS containing 10% FBS for 1 h at RT. Cells were incubated overnight at 4 °C with the primary antibodies. Astrocytes were identified using mouse-raised anti-GFAP (1:400, Millipore SAS, MAB360), and microglia were detected using rabbit-raised anti-Iba-1 (1:750, FUJIFILM Wako Pure Chemical Corporation, 019-19,741). On the following day, cells were washed with PBS containing 0.05% Tween-20 (PBS-T) (Sigma) and subsequently incubated at RT for 1 h with secondary antibodies conjugated to AlexaFluor 488 or AlexaFluor 568 fluorophores (1:400, all from Invitrogen, Waltham, MA, USA). Nuclei were stained with Hoechst 33,342 (1 μL/mL, Sigma) for 5 min. Coverslips were mounted in Mowiol (Sigma) over a microscope slide (Thermo Scientific, Waltham, MA, USA).

Immunostaining was visualized under an inverted widefield fluorescence microscope Zeiss Axiovert 200 (Carl Zeiss Inc., Berlin, Germany) using a 40× Plan-Apochromat (Zeiss, Germany) with a frame size of 1240 × 1240 pixels. The software AxioVision 4 (Carl Zeiss Inc.) was used for image acquisition. For each marker, images were acquired using the same exposure conditions. Astrocytes and microglia were determined by counting the number of GFAP-positive cells and Iba-1-positive cells, respectively, on four coverslips, five random fields in each coverslip, from 4 independent experiments.

### 2.8. Propidium Iodide Uptake Assay

Cell death was assessed by the cellular uptake of the fluorescence dye propidium iodide (PI, 3,8-diamino-5-(3-(diethylmethylamino) propyl)-6-phenyl phenanthridinium diiodide, Sigma). PI is a polar compound that interacts with DNA, emitting red fluorescence (630 nm; absorbance 493 nm). It is only permeable to compromised cell membranes; thus, only cells with damaged membranes can be identified with PI. PI is not toxic to live cells; therefore, it was used in immunocytochemistry assays to identify dead cells in the cultures.

PI staining was performed at 10 and 17 DIV. After a 20 min incubation period with PI (7.5 µM, PI stock solution at 1.5 mM) at 37 °C, 5% CO_2_ and 95% atmospheric air, an immunocytochemistry assay was conducted to identify GFAP-stained astrocytes. Immunostaining was visualized under an inverted widefield fluorescence microscope (Zeiss Axiovert 200), as described for the immunocytochemistry procedure (see Section 3.2.). Dead astrocytes were determined by counting the number of double-stained GFAP+ PI+ cells on four coverslips, five random fields in each coverslip, from 4 independent experiments, as described for the immunocytochemistry procedure.

### 2.9. Enzyme-Linked Immunosorbent Assay (ELISA)

Interleukin-1β (IL-1β) released to the culture medium was quantified by ELISA after the 24 h insult with microglia-derived factors. The assay was performed following the manufacturer’s protocol (DY501, R&D Systems, Minneapolis, MN, USA) using selective antibodies. Absorbance was read at 450 nm, with a 540 nm reference, in the Microplate Reader TECAN Infinite M200.

### 2.10. Statistical analysis

In this work, statistical significance was evaluated through GraphPad Prism, GraphPad Software (San Diego California, CA, USA). Data are expressed as mean ± standard error of the mean (SEM) from N independent experiments. Student’s *t*-test was used when comparing two conditions. One-way and two-way analysis of variance (ANOVA), followed by Dunnett’s and Sídák’s multiple comparison test, respectively, was used for comparing more than two conditions. For each data set, outliers were identified in GraphPad software through ROUT method with ROUT coefficient Q=1%. Statistical significance was considered when *p* < 0.05.

## 3. Results

### 3.1. PLX-3397 Efficiently Removes Microglia from Primary Mixed Glial Cultures

At 7 DIV, cultured forebrain astrocytes were trypsinized from T75 flasks and replated in culture dishes. After replating, to assess the efficacy of PLX-3397, cells were exposed to three different concentrations of the drug: 0.2 μM, 1 μM and 5 μM. Control cultures (CTL) were not exposed to PLX-3397. Regarding Iba-1 expression, Western blotting revealed a significant decrease with all concentrations of PLX-3397 tested—0.2 μM (0.626 ± 0.083; *p* < 0.01), 1 μM (0.472 ± 0.074; *p* < 0.01) and 5 μM (0.326 ± 0.051; *p* < 0.01)—in comparison with the control condition (CTL: 1.373 ± 0.092) (Figure 2A,B). None of the PLX-3397 concentrations affected GFAP expression (Figure 2A,C).

Using immunocytochemistry, it was also possible to observe the efficacy of PLX-3397 in the removal of microglia from primary cultures of astrocytes, thus corroborating the Western blot analysis. As depicted in Figure 3A, in all concentrations of PLX-3397 assessed, a visible reduction in Iba-1+ cells was observed. Indeed, a significant decrease in the percentage of Iba-1-expressing cells (0.2 μM: 1.847% ± 0.318, *p* < 0.0001; 1 µM: 0.923% ± 0.318, *p* < 0.0001; 5 µM: 0.700% ± 0.340, *p* < 0.0001 vs. CTL: 18.70% ± 3.425) and a significant increase in the percentage of GFAP-expressing cells (0.2 μM: 98.15% ± 0.318, *p* < 0.001; 1 µM: 99.08% ± 0.318, *p* < 0.0001; 5 µM: 99.30% ± 0.340, *p* < 0.0001 vs. CTL: 81.30% ± 3.425) were obtained by counting the number of Iba-1- and GFAP-stained cells at 17 DIV (Figure 3B).

### 3.2. PLX-3397 Has No Impact on Astrocyte Viability

We next addressed the impact of PLX-3397 on astrocyte viability. Cell viability was evaluated through a PI uptake assay at 10 and 17 DIV in control cultures and in cultures exposed to 0.2 µM, 1 µM and 5 µM of PLX-3397. At both time points, as depicted in Figure 4A, astrocytes exposed to PLX-3397 demonstrated similar PI uptake to control cultures. The immunocytochemistry assay was corroborated by the absence of a statistical difference (*p* > 0.05) in the number of GFAP+ PI+ cells counted in each condition when compared to cultures not exposed to PLX-3397 (10 DIV: 4.095% ± 0.693 and 17 DIV: 10.35% ± 0.580) (Figure 4B). Altogether, these results indicate that PLX-3397 is not toxic to astrocytes in the range of concentrations examined.

### 3.3. PLX-3397 Decreases Astrocytic Reactivity

Astrocytes are known to play an important role in the inflammatory response within the CNS. Through reactive astrogliosis, these cells react to damage/noxious insults and have the capability to adopt distinct states characterized by specific molecular profiles, specific functions, and distinct impacts on diseases. The spectrum of reactive astrocytes was recently considered to be between two extreme states: a neurotoxic state and a repair/neuroprotection-associated state [21]. These reactive states co-exist and constitute a heterogeneous population of reactive astrocytes [21,39,40]. Given the relevance of reactive astrogliosis in diseases, it was pertinent to assess whether PLX-3397 was affecting the reactive state of cultured astrocytes.

The complement system is a key innate immune defense against infection and an important driver of inflammation. Of these, Complement 3 (C3) is considered a marker of pro-inflammatory status, and it has become widely accepted that the neurotoxic phenotype of astrocytes can be identified by the upregulation of C3 [21,41,42]. Pentraxin 3 (PTX3) is also an essential component of the innate immune system, being an acute-phase protein that is produced rapidly at local sites of inflammation. Furthermore, a genomic study showed that the PTX3 gene was upregulated in neuroprotective astrocytes [41]. We thus decided to use C3 and PTX3 expression as indicators of astrocytic reactivity in PLX-3397-exposed cultures.

The expression of C3 and PTX3 was analyzed by Western blot in control cultures and in cultures exposed to 0.2 µM, 1 µM and 5 µM of PLX-3397. As depicted in Figure 5, PLX-3397-exposed cultures displayed a significant decrease in C3 (Figure 5B) (0.2 μM: 0.076 ± 0.030, *p* < 0.0001; 1 µM: 0.083 ± 0.032, *p* < 0.0001; 5 µM: 0.063 ± 0.022, *p* < 0.0001) when compared with control cultures (CTL: 0.802 ± 0.096). Likewise, a similar observation was made regarding PTX3 expression (Figure 5C) (0.2 μM: 0.523 ± 0.058, *p* < 0.001; 1 µM: 0.446 ± 0.054, *p* < 0.0001; 5 µM: 0.483 ± 0.042, *p* < 0.001 vs. CTL: 1.145 ± 0.186). These results reveal a decreased astrocytic reactivity in the presence of PLX-3397.

### 3.4. PLX-3397 Does Not Compromise Astrocytic Response 

Microglia and astrocytes tile the entire CNS, and their interaction affects neural cell functions in health and disease [43]. Signaling between microglia and astrocytes in the context of chronic neurodegenerative diseases drives many different transcriptomic, proteomic and functional responses in astrocytes.

Recently, it was found that reactive astrocytes were directly mediated by C1q complement, as well as Interleukin-1α (IL-1α) and Tumor Necrosis Factor-α (TNF-α), released by microglia [21]. Therefore, to determine whether PLX-3397-exposed astrocytes retain this property, 17-DIV astrocytes were stimulated with these microglia-derived factors (F: IL-1α (3 ng/mL), TNF-α (30 ng/mL) and C1q (400 ng/mL)) for 24 h, and the analysis of astroglial reactivity was accomplished.

In PLX-3397 (1 µM)-exposed cultures, F had no impact on Iba-1 (Figure 6A,B) or GFAP expression (Figure 6A,C) but upregulated C3 and PTX3 expression. In particular, F-stimulated astrocytes underwent a significant increase in C3 (Figure 6A,D) when compared with non-stimulated cultures (no stimulus: 0.037 ± 0.012, F: 0.456 ± 0.079, *p* < 0.01). The same observation was made for the expression of PTX3 (Figure 6A,E) (no stimulus: 0.070 ± 0.004, F: 0.943 ± 0.096, *p* < 0.0001).

To further corroborate the occurrence of reactive astrogliosis towards F, the release of IL-1β, a pro-inflammatory cytokine upregulated in inflammatory conditions, was quantified by ELISA. Indeed (Figure 6F), F induced the significant release of IL-1β to the culture medium (no stimulus: 80.11 ± 15.92; F: 241.1 ± 7.841, *p* < 0.001).

These results fully demonstrate that PLX-3397 does not compromise the immune response of cultured astrocytes.

## 4. Discussion

Over the years, astrocyte biology has received growing interest due to the prominent role of these cells in neuroinflammation. However, the presence of microglia in primary cultures makes it hard to clearly understand the precise role of astrocytes independently of microglia.

Microglia removal from mixed glial cultures has been a challenge, and with most methods so far reported, microglia still represent about 10% of the cell population in the culture, which is far too much due to their high reactivity potential towards inflammatory stimuli. The most widely used technique for achieving a single population of glial cells in primary cultures is based on their different adherence properties to the plastic of culture flasks/dishes [16]. This mechanical shake-off method allows the removal of microglial cells that are present above, but not below or inside, the astroglial monolayer, so the exclusion of microglial cells from mixed glial cultures is incomplete. The addition of anti-mitotic agents, such as 1-β-d-arabinofuranosylcytosine (Ara-C), was also explored to eliminate proliferating microglia from mixed cultures. Unfortunately, being a potent S-phase-specific anti-tumor agent, low concentrations of Ara-C induced growth arrest in astrocytes, while high concentrations caused cell death [22]. Others have used liposomal clodronate, a method based on the phagocytic capabilities of microglia [15,44,45]. Clodronate is a drug known to induce apoptosis in macrophages or macrophage-like cells [46] and can be delivered intracellularly through liposomes [47], since it is a hydrophilic molecule that cannot cross phospholipid membranes [48]. Kumamaru and coworkers [15] used liposome-encapsulated clodronate in mixed glial cultures to selectively induce cell death in phagocytic cells. These authors reported that liposomal clodronate was readily absorbed by microglia without affecting astrocytes. However, it is increasingly acknowledged that astrocytes have phagocytic abilities [49,50], which questions the effectiveness of liposomal clodronate addition. An antigen–antibody-mediated magnetic cell sorting system using CD11b MicroBeads antibody also allows for rapidly obtaining pure cultures of astrocytes after microglia depletion [20]. Complete microglia removal is accomplished, but this method is time-consuming and expensive. The most recent method described to purify astrocytes is immunopanning, which involves passing the cell suspension over a series of Petri dishes coated with antibodies directed against cell-type-specific antigens. In a typical immunopanning purification, a cell suspension from a nervous system region is prepared and then passed over the immunopanning dishes consecutively, with the first dish or two being used to deplete unwanted cell types and the final dish being used to select the cell type of interest, as described by Zhang and colleagues [23]. Purifying and culturing cells from the CNS by immunopanning is a powerful tool for dissecting fundamental neuron and glial properties and for understanding neuronal–glial interactions. It is a simple method, although laborious and costly, that provides high yield, but it takes practice, as every step must be performed correctly to still have viable cells by the end of the protocol.

Herein, we describe and validate a highly efficient way to eliminate microglia from primary cultures of astrocytes. Our approach is based on the inhibition of CSF-1R, a receptor crucial for microglia development and survival. No evidence of CSF-1R expression was found in neurons, astrocytes or neural precursor cells [51], corroborating its exclusive expression in microglia, as already stated by others [52,53]. Thus, we used a CSF-1R inhibitor, PLX-3397, and showed that it efficiently removes microglia from astrocytic cultures, therefore generating highly enriched monolayers of astrocytes.

PLX-3397 does not compromise astrocyte viability, and it reduces the expression of reactivity markers, such as C3 and PTX3. The fact that cultures not exposed to PLX-3397 had a higher expression of these proteins is most likely due to microglia that persisted after overnight shaking. These microglia can become activated due to shaking and not only express C3 and PTX3 but also release components that activate astrocytes. By eliminating these remaining reactive microglia, microglia-derived C3 and PTX3 and astrocyte stimulation are reduced, thus leading to low C3 and PTX3 expression. However, PLX-3397-exposed astrocytes still retain their responsiveness, as can be concluded by the increase in C3 and PTX3 expression and IL-1β release when enriched cultures were stimulated with microglia-derived factors. C3 upregulation and IL-1β release are associated with neurotoxic astrocytes, while increased PTX3 expression suggests the presence of the neuroprotective phenotype. These results corroborate the occurrence of a heterogeneous population of reactive astrocytes in the face of an insult, as recently claimed [39].

In conclusion, we prove that inhibiting CSF-1R via PLX-3397 is a simple and efficient approach to remove microglia from primary astroglial cultures. These enriched monolayers of astrocytes retain their immune properties, thus allowing for accurately understanding the individual response of astrocytes towards an inflammatory stimulus.

## Figures and Tables

**Figure 1 biomolecules-12-00666-f001:**
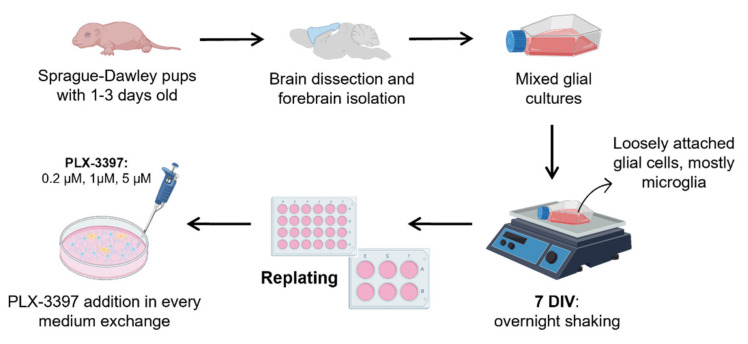
Step-by-step preparation of highly enriched cultures of astrocytes. Cells were maintained in T75 flasks from seeding until 7 DIV, when overnight shaking and replating took place. PLX-3397 was added from 7 DIV onwards in every medium exchange. DIV, days in vitro. Created with BioRender.com.

**Figure 2 biomolecules-12-00666-f002:**
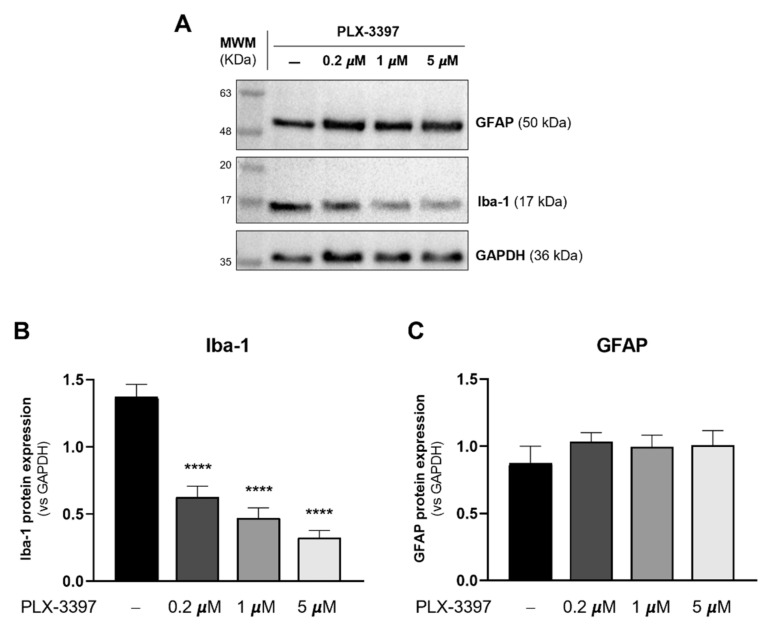
PLX-3397 decreases Iba-1 expression, but has no impact on GFAP expression, in primary cultures of astrocytes. (**A**) Representative immunoblots of GFAP, Iba-1 and GAPDH with no exposure (−) and exposure to 0.2 µm, 1 µm and 5 µM PLX-3397. Densitometric analysis of (**B**) Iba-1 and (**C**) GFAP was performed with ImageJ software using GAPDH as loading control. All values are mean ± SEM. N = 4 independent experiments. Statistical analysis was performed with one-way ANOVA followed by Dunnett’s multiple comparison test, **** *p* < 0.0001, in comparison with control condition.

**Figure 3 biomolecules-12-00666-f003:**
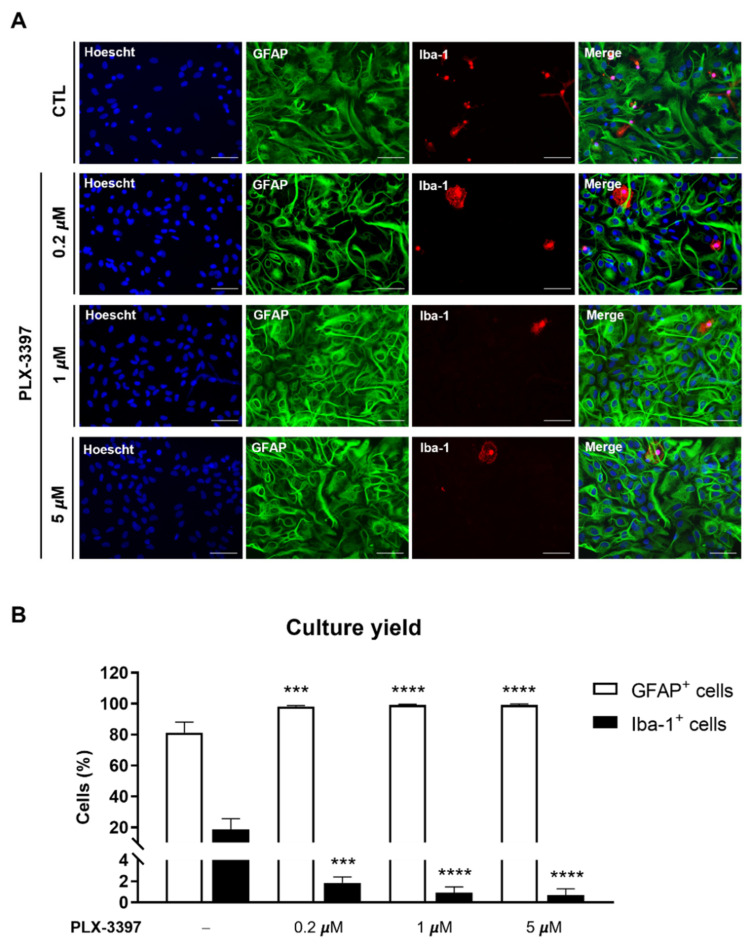
PLX-3397 significantly enhances culture yield. (**A**) Representative images of astrocytes and microglia in primary cultures of astrocytes with no exposure (CTL, −) and exposure to 0.2 µm, 1 µm and 5 µM of PLX-3397. GFAP-positive astrocytes are shown in green; microglia were identified with Iba-1 (red), and nuclei were counterstained with Hoechst (blue). Fluorescence images were acquired with a 40x objective in Zeiss Axiovert 200. Scale bar, 50 µm. (**B**) Cell count was performed in ImageJ software. All values are mean ± SEM. N ≈ 4500 cells per condition from 4 independent experiments. Statistical analysis was performed with two-way ANOVA followed by Sidak’s multiple comparison test, *** *p* < 0.001, **** *p* < 0.0001 in comparison with CTL condition.

**Figure 4 biomolecules-12-00666-f004:**
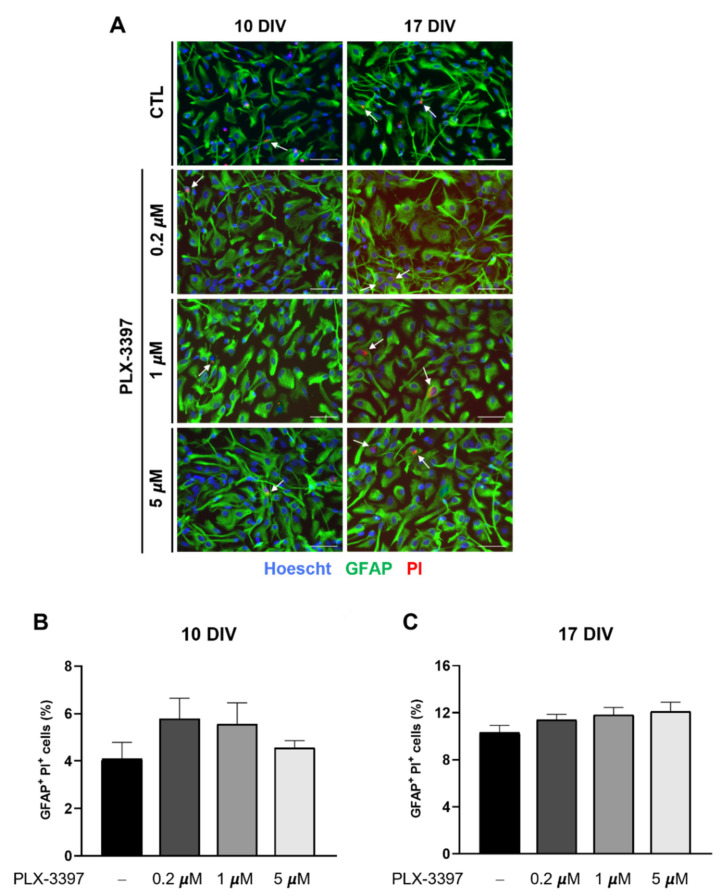
PLX-3397 does not compromise the viability of cultured astrocytes. (**A**) Representative images of astrocytes and PI staining and (**B**,**C**) number of double GFAP+ PI+ cells in primary cultures of astrocytes with no exposure (CTL, −) and exposure to 0.2 µm, 1 µm and 5 µM of PLX-3397 at 10 and 17 DIV. GFAP-positive astrocytes are shown in green, PI staining is red, and nuclei were counterstained with Hoechst (blue). Arrows point to PI labeling in astrocytes. Fluorescence images were acquired with a 40x objective in Zeiss Axiovert 200. Scale bar, 50 µm. Cell count was performed in ImageJ software. All values are mean ± SEM. N ≈ 3500 cells counted per condition at 10 DIV and N ≈ 5000 cells counted per condition at 17 DIV, from 4 independent experiments. Data are presented as a percentage of the total GFAP-positive cells. Statistical analysis was performed with one-way ANOVA followed by Dunnett’s multiple comparison test. DIV, days in vitro; PI, propidium iodide.

**Figure 5 biomolecules-12-00666-f005:**
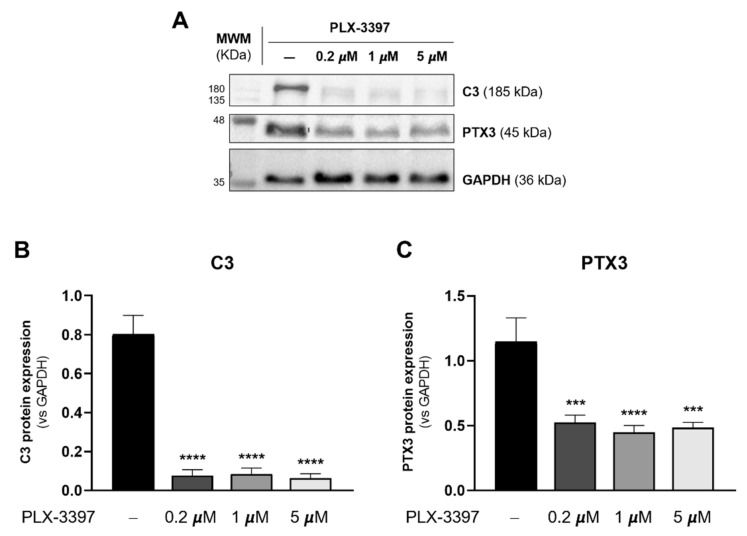
PLX-3397 decreases the reactivity of cultured astrocytes. (**A**) Representative immunoblots of C3, PTX3 and GAPDH with no exposure (−) and exposure to 0.2 µm, 1 µm and 5 µM of PLX-3397. Densitometric analysis of (**B**) C3 and (**C**) PTX3 was performed with ImageJ software using GAPDH as loading control. All values are mean ± SEM. N = 4 independent experiments. Statistical analysis was performed with one-way ANOVA followed by Dunnett’s multiple comparison test, *** *p* < 0.001, **** *p* < 0.0001 in comparison with control condition.

**Figure 6 biomolecules-12-00666-f006:**
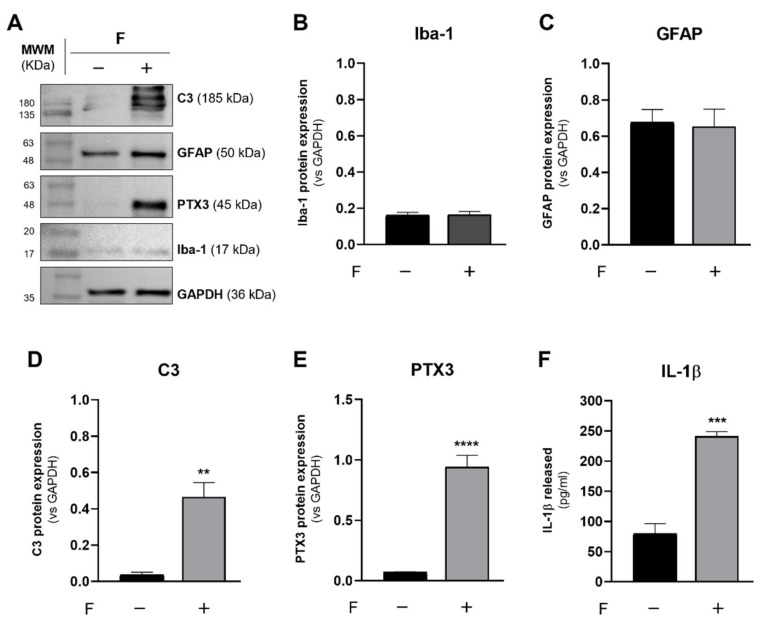
PLX-3397 does not impair astrocytic response induced by microglia-derived factors. Protein expression profile and IL-1β release in PLX-3397 (1 µM)-exposed cultures stimulated (+), or not (−), with microglia-derived factors (F: IL-1α, TNF-α and C1q). (**A**) Representative immunoblots of C3, PTX3, GFAP, Iba-1 and GAPDH. Densitometric analysis of (**B**) Iba-1, (**C**) GFAP, (**D**) C3 and (**E**) PTX3 was performed with ImageJ software using GAPDH as loading control. **(F)** IL-1β release to culture medium was quantified by ELISA. All values are mean ± SEM. N = 4 independent experiments. Statistical analysis was performed with Student’s *t*-test, ** *p* < 0.01, *** *p* < 0.001, **** *p* < 0.0001.

## Data Availability

The data presented in this study are available upon request to the corresponding author.

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
