# Peer review of "Microglia Depletion from Primary Glial Cultures Enables to Accurately Address the Immune Response of Astrocytes"

_biomolecules, 2022, doi:10.3390/biom12050666_

Round 1
Reviewer 1 Report
The authors present a method to increase the purification grade of astrocyte cultures by attempting to eliminate contamination microglia with the help of PLX-3397, a colony-stimulating factor-1 receptor inhibitor, which is exclusively expressed by microglia in CNS cultures, and which has been successful used previously in vivo. The study is well designed, documented and presented, but can be improved.
Major comment
The Introduction is far too long. I should be shortened by at least 50%. Par. on. p. 1, and par. 2 and 3 on p.3, should be the main objects for this action. Some of this information would be better to include in the Discussion, which is now fairly thin.
Author Response
The Introduction is far too long. It should be shortened by at least 50%. Par. on. p. 1, and par. 2 and 3 on p.3, should be the main objects for this action. Some of this information would be better to include in the Discussion, which is now fairly thin.
Reply: The authors have reorganized the Introduction and Discussion.

Reviewer 2 Report
In the present manuscript Van Zaller and co-authors evaluted the microglia depleting effect of the colony-stimulating factor-1 receptor (CSF-1R) inhibitor, PLX- 18 3397, in primary cultures of astrocytes. By performing Western blotting (Iba1, GFAP, C3, PTX3) and immunocytochemical analysis they investigated the dose-depending effects of 0.2-5uM PLX- 18 3397 on primary astrocyte viability and reactivity state.
They show that inhibiting CSF-1R, via PLX-3397, is a simple and efficient approach to remove microglia from primary astroglial cultures, without compromising astrocytes’ viability and turning them into a more resting-like state.
Although the manuscript might be of interest and provide some methodological implementation in performing microglia-depleting effects in vitro, some informations are lacking.
The authors tested the effect of PLX-3397 inhibitor in primary astrocyte cultures that were previously depleted of microglia by orbital shaking. Is the PLX-3397 inhibitor effective to deplete microglia in mixed glia cultures?
To test the efficiency of PLX-3397 and evaluate whether its use might be really effective to study inflammatory response in primary astrocyte cultures (as they claim), the authors should at least perturb the cellular system with an inflammatory insult and compare the response of astrocytes (and their secretome) when cultured in absence or in presence of PLX-3397.
Author Response
1. The authors tested the effect of PLX-3397 inhibitor in primary astrocyte cultures that were previously depleted of microglia by orbital shaking. Is the PLX-3397 inhibitor effective to deplete microglia in mixed glia cultures?
Reply: This brief report aimed to assess the efficacy of PLX-3397 combined with the method of overnight orbital shaking. To note that orbital shaking is widely used and considered essential to remove loosely attached cells from mixed cultures. Although microglia constitute the majority of these loosely attached cells, oligodendrocytes can also be present in mixed cultures being efficiently eliminated by shaking.
2. To test the efficiency of PLX-3397 and evaluate whether its use might be really effective to study inflammatory response in primary astrocyte cultures (as they claim), the authors should at least perturb the cellular system with an inflammatory insult and compare the response of astrocytes (and their secretome) when cultured in absence or in presence of PLX-3397.
Reply: The revised manuscript now includes Figure 6, which shows the response of the system towards a stimulus with microglia-derived factors (C1q, IL-1α, TNFα). Addition of microglia-derived factors induced the occurrence of inflammatory events, such as the upregulation of C3 and PTX3, together with an increase in IL-1b release to the culture medium, without affecting Iba-1 expression. This experiment proves that PLX-exposed astrocytes retain their immune properties and can thus be used to understand the individual response of astrocytes towards an inflammatory stimulus.
ELISA was described in Section Methods and Section Results now has one more subsection, named PLX-3397 does not compromise astrocytic response. The manuscript also has one more Figure, Figure 6.

Reviewer 3 Report
In vitro, cultured astrocytes have enormous usage in biological research as well as in pre-clinical studies. However, due to the microglial cross-contamination, many times researchers face significant hurdles in analyzing data. It has also been reported that contaminating microglia could change gene expression in astrocytes. Therefore, to eliminate microglia-induced unwarranted changes in astrocytes in vitro conditions, the authors have tested a small molecule, PLX-3397, that is known to deplete microglia in vivo. Overall, this study is well performed and important for the research community. Before acceptance, the authors should clarify the following points.
- In Fig. 2, the reduction in Iba-1 protein expression was approximately 60-70% in the PLX treated conditions. However, in Fig. 3, the number of Iba-1 positive cells in a similar setting was almost negligible (1-2%). The authors should clarify this disconnect in Iba-1 positive cells.
- In Fig. 3, Iba-1 positive cells look like clustering/enlarged in the presence of PLX. Please explain.
Author Response
1. In Fig. 2, the reduction in Iba-1 protein expression was approximately 60-70% in the PLX treated conditions. However, in Fig. 3, the number of Iba-1 positive cells in a similar setting was almost negligible (1-2%). The authors should clarify this disconnect in Iba-1 positive cells.
Reply: One possible explanation can be the different antibodies against Iba-1 used in western blot and immunocytochemistry assays. The authors used the antibodies according to the specifications. While goat anti-Iba-1 is suitable for western blot, rabbit anti-Iba-1 is considered adequate for immunohistochemistry assays.
2. In Fig. 3, Iba-1 positive cells look like clustering/enlarged in the presence of PLX. Please explain.
Reply: According to the literature, the colony-stimulating factor-1 receptor (CSF-1R) is expressed in microglia being crucial for their development and survival. By affecting microglia viability, CSF-1R inhibition through PLX-3397 will probably lead to changes in cellular morphology and phenotype. Enlarged microglia are indeed a characteristic of a reactive microglia state. However, this analysis was out of scope of the manuscript.

Reviewer 4 Report
The authors demonstrate that PLX-3397 mediated blockage of the CSF receptor can be used to remove microglia from primary isolated glial cultures. The description of the method is detailed and precise. For quantification of microglia and astrocyte content they perform WB and IF for IBA-1 and further characterize the effect of PLX-3397 on astrocytes by checking cell death with Pi and WB for C3 and PTX3.
In my opinion the following aspects would need further improvements or comments from the authors:
Introduction and discussion: Immunopanning should be mentioned as further purification method (e.g. PMID: 26687838 ) and advantages and disadvantages of the different purification methods should be discussed.
Lines 78, 85 replace "cell dead" with "cell death"
Lines158-160 “When dissolving a drug in DMSO, one must consider that, above a certain concentration, DMSO can be toxic to cells/tissues. However, the highest PLX-3397 concentration tested, 5 μm, corresponds to 0.005% of DMSO, which according to the literature is innocuous [43,44]. Thus, cultures not exposed to PLX-3397 were used as control.”
Why the authors did not treat the non purified cultures with DMSO only to have comparable conditions?
Lines 202-205“For that, after abundantly washed with TBST, membranes were incubated with Stripping Solution (1.5% glycine, 1% SDS, 203 10% Tween-20, H2O MQ and pH 2.2 corrected with acetic acid) for 30 min at RT. After stripping, membranes were abundantly washed with distilled water and several times with TBST. The protocol proceeded from the blocking step“
Did you confirm efficiency of your stripping step?
Lines 220-222“Astrocytes were identified using the mouse raised anti-glial fibrillary acidic protein antibody (anti-GFAP, 1:400, Millipore SAS, MAB360) and microglia were detected using the rabbit raised anti-ionized calcium-binding adaptor molecule 1 antibody (anti-Iba1, 1:750, FUJIFILM Wako Pure Chemical Corporation, 019-19741).”
Not all astrocytes need to be GFAP positive, even though this is the most widely used marker. Did the authors consider staining for further contaminants (NSC/Radial glia/OPC/Oligodendrocytes and neurons)? Overview pictures with lower magnification might help to judge the overall purity of the culture
Lines 231-232 For each marker, images were acquired under identical lighting between conditions.
I would suggest: "Using the same exposure conditions"
Figure 2 and 3: In figure 3 the authors show a reduction of IBA1 positive cells from approx 20% to 1% upon treatment with PLX-3397. However on the protein level IBA1 reduces only about 50% in Fig. 2 possible reasons for this discrepancy should be discussed.
In Figure Fig3a the authors show the effect of PLX-3397 on the primary cultures using Immunoflourescence stainigs for GFAP and IBA1. IBA-1 positive cells vanish with the PLX-3397 treatment. However, the GFAP stain and morphology of the astrocytes seems also to change upon PLX-3397 treatment from a ramified morphology and a homogenous cytoplasmic distribution to a more bipolar cell with a stronger and more bundled staining pattern. This point to a form of glial activation, either through dying microglia or the PLX-3397 itself
Fig5. The authors take C3 and PTX3 expression as surrogates for assessing astrocyte activation. However, both of these genes are also known to be expressed in microglia. Without careful assessment what the astrocytic and microglia contribution is in the mixed cultures, any statement on the astrocytic phenotype using those two proteins is speculative and does not contribute to the value of the ms.
While the method description is of high quality, assessment of astrocytic activation was performed with conceptional errors and would need significantly more experimental efforts.
Author Response
1. Introduction and discussion: Immunopanning should be mentioned as further purification method (e.g. PMID: 26687838) and advantages and disadvantages of the different purification methods should be discussed.
Reply: Immunopanning is mentioned in the revised version of the manuscript. Advantages and disadvantages of the different purification methods are now discussed. Zhang et al., 2016 was added to references.
2. Lines 78, 85 replace "cell dead" with "cell death"
Reply: The alteration was performed.
3. Lines158-160 “When dissolving a drug in DMSO, one must consider that, above a certain concentration, DMSO can be toxic to cells/tissues. However, the highest PLX-3397 concentration tested, 5 μm, corresponds to 0.005% of DMSO, which according to the literature is innocuous [43,44]. Thus, cultures not exposed to PLX-3397 were used as control.” Why the authors did not treat the non purified cultures with DMSO only to have comparable conditions?
Reply: As the authors mentioned in the manuscript, the highest PLX-3397 concentration tested, 5 μm, corresponds to 0.005% of DMSO, which according to the literature is innocuous. Therefore, DMSO impact was considered negligible, and the authors didn’t add it in non-purified cultures.
4. Lines 202-205 “For that, after abundantly washed with TBST, membranes were incubated with Stripping Solution (1.5% glycine, 1% SDS, 203 10% Tween-20, H2O MQ and pH 2.2 corrected with acetic acid) for 30 min at RT. After stripping, membranes were abundantly washed with distilled water and several times with TBST. The protocol proceeded from the blocking step“. Did you confirm efficiency of your stripping step?
Reply: The stripping step is used by many researchers worldwide. The authors used a mild stripping solution (https://www.abcam.com/ps/pdf/protocols/stripping%20for%20reprobing.pdf), suggested by Abcam, a renowned company in the field of antibodies. In the past, the authors have checked the efficiency of this stripping solution by incubating the PVDF membrane with chemiluminescence detection reagent after stripping and found that stripping was always highly efficient.
5. Lines 220-222 “Astrocytes were identified using the mouse raised anti-glial fibrillary acidic protein antibody (anti-GFAP, 1:400, Millipore SAS, MAB360) and microglia were detected using the rabbit raised anti-ionized calcium-binding adaptor molecule 1 antibody (anti-Iba1, 1:750, FUJIFILM Wako Pure Chemical Corporation, 019-19741).” Not all astrocytes need to be GFAP positive, even though this is the most widely used marker. Did the authors consider staining for further contaminants (NSC/Radial glia/OPC/Oligodendrocytes and neurons)? Overview pictures with lower magnification might help to judge the overall purity of the culture.
Reply: The method used to prepare the cultures originates mainly mixed cultures of astrocytes and microglia, as described by many researchers in the field. To prepare NSC/Radial glia/OPC/Oligodendrocyte/neurons other protocols, which use animals with other ages and more suitable culture media, are recommended. With the protocol described in the manuscript these other cell types are negligible, if present at all, and the authors didn’t consider necessary to look for other markers.
6. Lines 231-232 For each marker, images were acquired under identical lighting between conditions. I would suggest: "Using the same exposure conditions"
Reply: The authors addressed this comment.
7. Figure 2 and 3: In figure 3 the authors show a reduction of IBA1 positive cells from approx 20% to 1% upon treatment with PLX-3397. However on the protein level IBA1 reduces only about 50% in Fig. 2 possible reasons for this discrepancy should be discussed.
Reply: One possible explanation can be the different antibodies against Iba-1 used in western blot and immunocytochemistry assays. The authors used the antibodies according to the specifications. While goat anti-Iba-1 is suitable for western blot, rabbit anti-Iba-1 is considered adequate for immunohistochemistry assays.
8. In Fig. 3a the authors show the effect of PLX-3397 on the primary cultures using Immunoflourescence stainigs for GFAP and IBA1. IBA-1 positive cells vanish with the PLX-3397 treatment. However, the GFAP stain and morphology of the astrocytes seems also to change upon PLX-3397 treatment from a ramified morphology and a homogenous cytoplasmic distribution to a more bipolar cell with a stronger and more bundled staining pattern. This point to a form of glial activation, either through dying microglia or the PLX-3397 itself.
Reply: GFAP is considered a good marker for astrocytes, but the concept that changes in GFAP expression are associated with astrogliosis is no longer considered accurate. Indeed, Escartin and co-workers (Ref. 39 in the manuscript) have reviewed this issue and stated that changes in GFAP expression may also reflect physiological adaptive plasticity rather than being simply a reactive response to stimuli. Morphology of astrocytes and GFAP expression are no longer sufficient criteria to qualify astrocytes as reactive.
9. In Fig. 5 the authors take C3 and PTX3 expression as surrogates for assessing astrocyte activation. However, both of these genes are also known to be expressed in microglia. Without careful assessment what the astrocytic and microglia contribution is in the mixed cultures, any statement on the astrocytic phenotype using those two proteins is speculative and does not contribute to the value of the ms.
10. While the method description is of high quality, assessment of astrocytic activation was performed with conceptional errors and would need significantly more experimental efforts.
Reply to comments 9 and 10: Indeed, both C3 and PTX3 are essential components of the innate immune system, therefore expressed in immune cells as microglia. However, these proteins were closely associated to reactive astrogliosis. C3 upregulation in astrocytes was observed in many brain diseases (Liddelow et al., 2017) and several recent publications use C3 as a marker for neurotoxic astrocytes (Ref. 42 - Grimaldi et al., 2019). In what concerns PTX3, a genomic study identified it as a marker of neuroprotective astrocytes (Zamanian et al., 2012).
As such, and given the negligible presence of microglia in PLX-exposed cultures, the authors decided to use C3 and PTX3 as indicators of reactive astrogliosis, and thus, of astrocytic immune response.
The revised manuscript now includes Figure 6, which shows the response of the system towards a stimulus with microglia-derived factors (C1q, IL-1α, TNFα). Addition of microglia-derived factors induced the occurrence of inflammatory events, such as the upregulation of C3 and PTX3, together with an increase in IL-1b release to the culture medium, without affecting Iba-1 expression. The increase in C3 and PTX3 can only derive from reactive astrocytes, proving that in this system, as in many others, C3 and PTX3 are indeed markers of reactive astrogliosis. This experiment also proves that PLX-exposed astrocytes retain their immune properties and can thus be used to understand the individual response of astrocytes towards an inflammatory stimulus.
ELISA was described in Section Methods and Section Results now has one more subsection, named PLX-3397 does not compromise astrocytic response. The manuscript also has one more Figure, Figure 6.

Round 2
Reviewer 1 Report
The authors have revised the manuscript according to recommendations.
Author Response
Thank you for your review.
Reviewer 2 Report
The authors performed the experiments required.
Author Response
Thank you for your review.